# Synergistic Activity of Pep16, a Promising New Antibacterial Pseudopeptide against Multidrug-Resistant Organisms, in Combination with Colistin against Multidrug-Resistant *Escherichia coli*, In Vitro and in a Murine Peritonitis Model

**DOI:** 10.3390/antibiotics12010081

**Published:** 2023-01-03

**Authors:** Samuel Chosidow, Bruno Fantin, Irène Nicolas, Jean-Baptiste Mascary, Françoise Chau, Valérie Bordeau, Marie-Clemence Verdier, Pierre Rocheteau, Francois Guérin, Vincent Cattoir, Victoire de Lastours

**Affiliations:** 1IAME UMR-1137, INSERM, Université de Paris, F-75018 Paris, France; 2Service de Médecine Interne, Hôpital Beaujon, AP-HP, F-92210 Clichy, France; 3SAS. Olgram, F-56580 Bréhan, France; 4Unité Inserm U1230 BRM, Université de Rennes 1, F-35043 Rennes, France; 5Laboratoire de Pharmacologie Biologique, CHU Pontchaillou, F-35033 Rennes, France; 6Service de Bactériologie-Hygiène Hospitalière & CNR de la Résistance aux Antibiotiques (Laboratoire Associé “Entérocoques”), CHU Pontchaillou, F-35033 Rennes, France

**Keywords:** *Escherichia coli*, multiresistance, antimicrobial peptides, colistin, murine models

## Abstract

Colistin is a drug of last resort to treat extreme drug-resistant Enterobacterales, but is limited by dose-dependent toxicity and the emergence of resistance. A recently developed antimicrobial pseudopeptide, Pep16, which acts on the cell membrane, may be synergistic with colistin and limit the emergence of resistance. We investigated Pep16 activity against *Escherichia coli* with varying susceptibility to colistin, in vitro and in a murine peritonitis model. Two isogenic derivatives of *E. coli* CFT073 (susceptible and resistant to colistin) and 2 clinical isolates (susceptible (B119) and resistant to colistin (Af31)) were used. Pep16 activity, alone and in combination with colistin, was determined in vitro (checkerboard experiments, time–kill curves, and flow cytometry to investigate membrane permeability). Toxicity and pharmacokinetic analyses of subcutaneous Pep16 were performed in mice, followed by the investigation of 10 mg/kg Pep16 + 10 mg/kg colistin (mimicking human concentrations) in a murine peritonitis model. Pep16 alone was inactive (MICs = 32–64 mg/L; no bactericidal effect). A concentration-dependent bactericidal synergy of Pep16 with colistin was evidenced on all strains, confirmed by flow cytometry. In vivo, Pep16 alone was ineffective. When Pep16 and colistin were combined, a significant decrease in bacterial counts in the spleen was evidenced, and the combination prevented the emergence of colistin-resistant mutants, compared to colistin alone. Pep16 synergizes with colistin in vitro, and the combination is more effective than colistin alone in a murine peritonitis by reducing bacterial counts and the emergence of resistance. Pep16 may optimize colistin use, by decreasing the doses needed, while limiting the emergence of colistin-resistant mutants.

## 1. Introduction 

The emergence and subsequent worldwide diffusion of extended-spectrum β-lactamase-producing Enterobacteriaceae (ESBL-Es), which are resistant to third-generation cephalosporins, has led to a dramatic increase in carbapenem consumption [1]. In this context, carbapenemase-producing *Enterobacterales* (CPEs) have emerged and spread globally. Notably, metallobetalactamases (MBLs) generate resistance to most available drugs, including the newly developed combinations of cephalosporins with β-lactamase inhibitors, such as ceftazidime–avibactam and meropenem–varbobactam, or the new aminoglycoside, plazomicin. This prompts the use of last-resort antibiotics, such as colistin, a polypeptide antibiotic, which remains active on 80% of CPEs including MBLs [2]. However, the use of this “old antibiotic” has been limited by its dose-dependent nephrotoxicity, its high inoculum effect, and its high risk of selecting resistant mutants in vivo [3,4,5]. Strains carrying resistance to colistin have subsequently appeared, leading to situations where few, if any, therapeutic options remain [6]. Additionally, the global spread of plasmids carrying *mcr*-like genes that confer colistin resistance could impede its further use [7,8]. Moreover, co-production of MCR-1 and MBLs by *E. coli* has recently been described [9]. Developing alternative antimicrobials that remain active against these highly resistant strains is a public health priority [10]. In the meantime, available clinical data suggest that combinations of “old” antibiotics are more effective in the management of infections caused by CPEs [11,12]. In vitro data also showed synergy between colistin and other antibiotics, even against colistin-resistant strains. For instance, we have recently shown that the combination of colistin and fosfomycin was beneficial in vitro and in vivo against NDM-1-producing *E. coli*, even with strains less susceptible to colistin and fosfomycin [13].

Recently, antimicrobial peptides inspired by and imitating a section of a *Staphylococcus aureus* toxin have been synthesized [14]. They have shown bacteriostatic and bactericidal effects against several Gram-positive and -negative multidrug-resistant (MDR) bacteria by altering membrane permeability, leading to bacterial death. One of these peptides, Pep16, has been studied in a murine model of sepsis due to methicillin-resistant *S. aureus* and succeeded in reducing 48 h mortality. This work also demonstrated that the use of Pep16 was associated with a low toxicity and low emergence of resistance, two imperative qualities of an antimicrobial drug candidate. Because of the urgent need for new antibiotic compounds to treat infections caused by MDR Enterobacteriaceae, and CPEs especially, we tested here the efficacy of Pep16 on *Escherichia coli* infections, as an update to the first paper published in 2019 by Nicolas et al. [14]. 

We hypothesized that combining Pep16 and colistin, which both operate on the outer membrane of Gram-negative bacteria, may potentiate their antibacterial effect. This may allow for the possibility of using lower, less-toxic doses, the decrease in the selection of resistance, or even the restoration of colistin activity in case of resistance. We developed a high-inoculum severe murine peritonitis model, responsible for 100% mice mortality in the absence of treatment. This model allows testing for animal survival and bacterial counts in the spleen and organ sterilization and is especially interesting to test for the efficacy of combinations of antibacterials [13,15,16,17]. 

Thus, the aim of this study was to determine the activity of Pep16 in vitro and in a murine model of peritonitis due to *E. coli*, alone and in combination with colistin. 

## 2. Methods

### 2.1. Bacterial Strains

Four bacterial strains were used: (i) the plasmid pCR-Blunt II-TOPO (Life Technologies, Saint-Aubin, France), which carries a kanamycin resistance gene, was electroporated in an uropathogenic strain, *E. coli* CFT073, to construct *E. coli* CFT073-pTOPO, a strain previously used by our team in the peritonitis model [15,17]; (ii) a colistin-resistant isogenic strain *E. coli* CFT073-pTOPO-COLR previously selected under treatment with colistin in the peritoneal fluid from mice using the same murine peritonitis model [3]; (iii) two clinical isolates, one susceptible strain (*E. coli* B119) and one strain harboring the *mcr-1*-positive plasmid and resistant to colistin (*E. coli* Af31).

### 2.2. Antibiotics

The antibiotics used were colistin sulfate (Sigma-Aldrich, Saint-Quentin-Fallavier, France) and cefotaxime (Sigma-Aldrich, Saint-Quentin-Fallavier, France for in vitro experiments and Mylan, Saint-Priest, France for in vivo experiments). Pep16 was supplied by S.A.S. Olgram (Bréhan, France), the manufacturer of the drug, with whom this work was conducted in collaboration. Details on the synthesization of Pep16, inspired by and imitating a section of a *Staphylococcus aureus* toxin, can be found in the princeps paper published in 2019 [14].

### 2.3. In Vitro Experiments

#### 2.3.1. MICs

The MICs of Pep16 and colistin were determined by the broth microdilution method in accordance with the EUCAST guidelines (www.eucast.org, accessed on 2 September 2022). In order to investigate for the presence of a potential inoculum effect in vitro, MICs were performed with various inoculum sizes, increasing from 10^5^ to 10^7^ CFU/mL. To explore MIC variations in the presence of albumin, MICs were also determined in Mueller–Hinton broth (MHB) supplemented with 4% human albumin.

#### 2.3.2. Bacteriostatic and Bactericidal Synergy Tests

The existence of an in vitro synergy between colistin and Pep16 was tested by checkerboard synergy testing and time–kill curves using both antibiotics separately and in combination. Checkerboard synergy testing was performed by the broth microdilution method [18]. Combinations of colistin and Pep16 were tested at concentrations of 0.03–32 and 1–64 mg/L, respectively. The fractional inhibitory concentration (FIC) index was calculated by adding the FICs (the MIC of Drug A in combination with Drug B over the MIC of Drug A alone) of colistin and Pep16. An FIC index < 0.5 defined synergy.

Time–kill curves were performed with Pep16 (4–128 mg/L) and colistin (0.5–8 mg/L) alone or in combination. Exponentially growing *E. coli* cells were incubated in 10 mL of Mueller–Hinton broth to obtain 10^6^ CFU/mL with Pep16 and/or colistin. Viable counts were enumerated by plating 100 μL of appropriate culture dilutions on to lysogeny broth (LB) agar plates after 0, 1, 3, 6, and 24 h of incubation at 37 °C and expressed in log_10_ CFU/mL. The lower limit of detection was 1 log_10_ CFU/mL. A bactericidal effect is defined as a 3 log_10_ decrease in CFU counts compared with the initial inoculum. A synergistic effect is defined as a 2 log_10_ decrease in CFU counts between the combination and its most-active constituent after 24 h [19]. Additionally, the number of surviving organisms in the presence of the combination had to be 2 log_10_ CFU/mL below the starting inoculum. The MICs of colistin-resistant mutants after 24 h of colistin monotherapy were determined (see above). All these experiments were repeated at least three times.

Flow cytometry was used to investigate membrane permeability after exposure to antibiotics. Experiments were executed on a Fortessa X-20 flow cytometer (Becton Dickinson, Le Pont de Claix, France) equipped with a 488 nm laser. Exponentially growing *E. coli* CFT073-pTOPO cells at 10^6^ CFU/mL were obtained and incubated with colistin and/or Pep16 at 0.25× MIC during 30 min. A negative control was incubated without antibiotics, and a positive control was incubated with a solution of 90% ethanol. The samples were centrifuged and reconstituted in phosphate-buffered saline twice. Cells were then stained by using a Live/Dead BacLight kit (Molecular Probes, Invitrogen, Cergy-Pontoise, France) as previously shown [20]. The kit consists of two fluorescent intercalating agents, propidium iodide (PI) and SYTO9, which both stain nucleic acids. The green-fluorescing SYTO9 is able to enter all cells when used alone, whereas the red-fluorescing PI only enters cells with damaged cytoplasmic membranes. The appropriate mixture of the SYTO9 and PI stains enables differentiation between bacteria with intact cytoplasmic membranes and dead bacteria with permeabilized cytoplasmic membranes. Optical filters were set up such that red fluorescence was measured at 635 nm and green fluorescence was measured at 500 nm. The proportion of dead cells (colored in red)/marked cells (colored in red and/or green) was retrieved for each condition. A minimum of 50,000 events per condition were recorded.

### 2.4. In Vivo Experiments

#### 2.4.1. Ethics

Animal experiments were performed in accordance with the prevailing regulations and ethical requirements of the Direction of Veterinary Services of Paris. The experimental protocol was approved by the French Ministry of Research and by the Ethics Committee for Animal Experiments (No. APAFIS 22330-2019092415325730). Animals were housed in regulation cages and given free access to food and water. Swiss ICR-strain female mice weighing 25 to 30 g were used.

#### 2.4.2. Survival Study

Non-infected mice survival was assessed after a single subcutaneous administration of Pep16 at 10 or 100 mg/kg. Mice were monitored for a minimum of 6 h after administration of Pep16.

#### 2.4.3. Pep16 Pharmacokinetic Analysis

Pep16 concentrations were determined in plasma from non-infected mice 0.5, 1, 3, 6, 12, and 24 h after subcutaneous Pep16 injection. Blood samples of 40–300 µL were collected in Eppendorf Protein LoBind Tubes (Sigma-Aldrich, Saint-Quentin-Fallavier, France) through intracardiac puncture. Tubes were then centrifuged for 5 min at 8000 rpm, and the plasma supernatant was retrieved and stored at −80 °C. Dosages were then performed by liquid chromatography-tandem mass spectrometry (LC/MS). Samples were prepared for analysis by adding 30 µL of analyte to 30 µL of internal standard, 30 µL of water/methanol (*v/v* 50/50), and 90 µL of 2% ZnSO_4_ acetonitril solution. Daptomycin was used as the internal control. The mix was then vortexed for 10 min at 10,000 rpm. Then, 100 µL of the supernatant was transferred into vials containing 900 µL of water/methanol (50/50, *v/v*), and 2 µL was then injected into the chromatographic system. A HYPERSIL GOLD™ C18 (ThermoFisher Scientific, Waltham, MA, USA) was used to perform chromatographic separation of Pep16 from the sample. The mobile phase used was constituted of water with 0.1% formic acid (A) and acetonitril with 0.1% formic acid (B). This column was maintained at 40 °C. The gradient started with 10%(B) and ended with 90% (B) for 7 min. The chromatography system was coupled to a triple-quadrupole Xevo TQ-XS^®^ mass spectrometer (Wasters, Milford, MA, USA). The method has a fixation concentration range going from 1 to 120 µg/mL.

#### 2.4.4. Murine Peritoneal Infection Model 

We used the lethal murine intraperitoneal infection model previously developed by our group [13,15,16,17]. Mice were inoculated by the intraperitoneal route with 250 μL of a bacterial suspension in porcine mucin 10% (Sigma–Aldrich), corresponding to a final inoculum of approximately 2.5 × 10^7^ CFU/mL. The strains tested were CFT073-pTOPO and CFT073-pTOPO-COLR. Two hours after inoculation, at least three mice per strain were sacrificed to determine the pre-therapeutic bacterial loads, referred to as start-of-treatment controls, since all untreated mice died within 12 h (data not shown), preventing the use of end-of-treatment controls. Each treatment group was composed of 9 mice, which were administered a single dose of antibiotic (or of a combination of antibiotics) subcutaneously 2 h after inoculation. The treatment groups were Pep16 10 mg/kg, colistin 10 mg/kg, cefotaxime 100 mg/kg, and Pep16 10 mg/kg + colistin 10 mg/kg. Doses of cefotaxime and colistin were chosen based on previous pharmacokinetic analysis, while the dose of Pep16 was chosen based on the present survival study and pharmacokinetic analysis. In each treatment group, mice were sacrificed by 200 µL of pentobarbital 40% (Euthasol VetVR, Dechra Veterinary Products, Montigny-Le-Bretonneux, France) in a sequential way: 3 mice 3 h after treatment, 3 mice 6 h after treatment, and 3 mice 12 h after treatment. Immediately after death, the spleens were extracted and homogenized. Samples were plated on LB agar. To detect resistant mutants after treatment with colistin, samples were also plated on LB agar containing colistin at 4× MIC. The MICs of colistin-resistant mutants were determined (see above). Colony counts were then determined after 48 h of culture. Endpoints were spontaneous death (before the planned time of the sacrifice), bacterial counts in the spleen, and the emergence of colistin-resistant mutants. The assessment criteria of the murine model were: (i) mortality rate; (ii) bacterial counts in peritoneal fluid (PF) and the spleen; and (iii) the selection of colistin-resistant mutants after 24 h of treatment.

### 2.5. Statistical Analysis

Continuous variables are expressed as the median and ranges (minimum to maximum) and compared using the non-parametric Kruskal–Wallis test followed, when significant, by the Mann–Whitney test for comparisons between two groups. Proportions were compared using Fisher exact or Chi-squared tests, when appropriate. Statistical analyses were performed with the R software (version 3.6.0). A *p*-value < 0.05 was considered significant.

## 3. Results

### 3.1. In Vitro Studies

#### 3.1.1. Minimum Inhibitory Concentrations

The MICs of Pep16 and colistin for the studied strains of *E. coli* are shown in Table 1. Increases in the inoculum of CFT073-pTOPO from 10^5^ CFU/mL to 10^6^ CFU/mL and 10^7^ CFU/mL were associated with an increase in the MICs of Pep16 from 64 mg/L to 128 mg/L and 1024 mg/L, respectively. The MICs of Pep16 for CFT073-pTOPO in the presence of 4% human albumin increased from 64 to >128 mg/L.

#### 3.1.2. Synergy Studies

The checkerboard method showed a synergistic interaction between Pep16 and colistin on the two colistin-susceptible *E. coli* strains CFT073-pTOPO and B119 with median FIC indices of 0.19 and 0.28, respectively. For both colistin-resistant strains, median FIC indices were above 0.5, indicating the absence of synergy. Using time–kill curves, a synergistic effect was observed with the combination of Pep16 and colistin against all studied strains, as shown in Figure 1. When used alone, Pep16 did not show any bactericidal effect, despite high concentrations (Panel A in Appendix A). The minimal concentrations that yielded a synergy for CFT073-pTOPO were Pep16 16 mg/L (0.25× MIC) and colistin 2 mg/L (2× MIC) (Figure 1). For both susceptible strains CFT073-pTOPO and B119, the combination of both molecules prevented the emergence of colistin-resistant mutants that was generated by the exposure to colistin alone (MIC of colistin-resistant mutants: 8–16 mg/L). When used alone, high concentrations of colistin were needed to impede the emergence of mutants (Panel B in Appendix A). Notably, the concentrations needed to show synergy with the colistin-resistant strains were significantly higher: Pep16 64 mg/L (2× MIC and 1× MIC for Af31 and CFT073-pTOPO-COLR, respectively) and colistin 8 mg/L (0.5× MIC and 1× MIC for Af31 and CFT073-pTOPO-COLR, respectively).

Figure 2 shows the results of the flow cytometry experiments used to investigate membrane permeability after exposure to antibiotics. In the study of membrane permeability on CFT073-pTOPO, the percentage of dead cells/marked cells after exposure to Pep16 0.25× MIC, to colistin 0.25× MIC and to the combination of both molecules, was 2%, 60%, and 82% (*p* < 0.001), respectively (Figure 2). In the absence of a validated definition of synergy in flow cytometry experiments, the increase in the percentage of dead cells/marked cells with the combination of Pep16 and colistin suggests a beneficial effect of the combination against the outer bacterial cell membrane. 

### 3.2. In Vivo Studies

#### 3.2.1. Toxicity Study

A total of 36 mice were treated with a single dose of Pep16, and six were used as controls and sacrificed 2 h after inoculation. Among the 18 mice who received Pep16 at a dose of 100 mg/kg subcutaneously, 6 (33%) survived at least 6 h. At a dose of 10 mg/kg of Pep16 administered subcutaneously, all (18/18) survived at least 6 h, including six mice that were monitored for 24 h.

#### 3.2.2. Pep16 Pharmacokinetic Analysis

As shown in Figure 3A, the pharmacokinetic analysis of Pep16 after a single subcutaneous administration of 10 mg/kg showed a slow increase in plasma concentrations, with a Cmax of #3 mg/L and a Tmax of 6 h. The full data are available in Appendix A (as Appendix A). Of note, pharmacokinetic analysis of colistin and cefotaxime had already been completed and was not repeated for ethical reasons [3,21].

#### 3.2.3. Murine Peritoneal Infection Model—In Vivo Time–Kill Curves

First, the susceptible strain *E. coli* CFT073-pTOPO was used to create the murine peritoneal infection model. In groups of infected mice treated in monotherapy with either Pep16 10 mg/kg or colistin 10 mg/kg, 6/8 (75%) mice and 1/9 (11%) mice died before the planned time of the sacrifice, respectively, as compared with no spontaneous death in mice treated with the antibiotic combination (0/9 mice) or for the group treated with the reference treatment cefotaxime 100 mg/kg (0/9 mice) (*p* < 0.001). The mean bacterial counts in the spleen were 7.7 × 10^8^ CFU/g, 2.7 × 10^8^ CFU/g, 1.6 × 10^7^ CFU/g and 3.6 × 10^6^ CFU/g in the groups treated with Pep16 10 mg/kg, colistin 10 mg/kg, colistin 10 mg/kg + Pep16 10 mg/kg, and cefotaxime 100 mg/kg, respectively (*p* = 0.002 for comparison of the four treatment groups). In particular, bacterial counts in the spleen of mice treated with the combination of colistin 10 mg/kg + Pep16 10 mg/kg were significantly lower than those from mice treated with colistin alone (*p* = 0.017) (Figure 3B) and Pep16 alone (*p* = 0.004). The mean counts of colistin-resistant mutants were significantly higher in the spleens of mice treated with colistin alone compared to those treated with the combination of colistin and Pep16 (1.6 × 10^3^ CFU/g vs. 2.3 × 10^1^ CFU/g, respectively; *p* < 0.001), as well as the proportion of mice with colistin-resistant mutants (9/9 vs. 3/9, respectively; *p* = 0.01). The MICs of colistin-resistant mutants ranged from 8 to 16 mg/L.

Second, *E. coli* CFT073-pTOPO-COLR was used in the murine peritoneal infection model. In the groups treated with Pep16 10 mg/kg, colistin 10 mg/kg, or colistin 10 mg/kg + Pep16 10 mg/kg, 3/7 mice, 5/9 mice and 5/9 mice died before the planned time of the sacrifice, respectively (*p* = 0.84), and the mean bacterial counts in the spleen were 3.3 × 10^8^ CFU/g, 4.0 × 10^8^ CFU/g and 5.4 × 10^8^ CFU/g, respectively (*p* = 0.09).

## 4. Discussion

Despite high MICs and microbiological inefficiency when used alone, Pep16 is synergistic with colistin in vitro, and the combination is beneficial in vivo, even at sub-inhibitory concentrations, against colistin-susceptible isolates, and limits the emergence of colistin-resistant mutants. Although disappointing concerning the potential bactericidal effect of Pep16 alone, these findings are important as using Pep16 with colistin could limit the doses of colistin needed to obtain efficacy while reducing the emergence of colistin-resistant mutants.

In vitro, Pep16 alone did not show any activity on the different *E. coli* strains, with high MICs and no bactericidal effect even at high concentrations. The increase in the MICs in the presence of 4% human albumin indicates a high protein-bound fraction, and Pep16 showed a significant inoculum effect, albeit lower than colistin and cefotaxime. In addition to these findings, Pep16, when administered alone in the murine peritoneal infection model at 10 mg/kg, showed neither efficacy on spontaneous mortality nor on bacterial counts in the spleen. Owing to the high toxicity at 100 mg/kg, which was evidenced in the survival study, increasing the dose above 10 mg/kg was not deemed relevant.

In vitro, the combination of Pep16 with colistin proved to be synergistic. It allowed the reduction in colistin concentrations needed to reach a bacteriostatic effect on colistin-susceptible strains and a bactericidal effect on colistin-susceptible and -resistant strains. Notably, the combination of these molecules avoided the emergence of colistin-resistant mutants, a common and important limitation to the use of colistin in clinical practice [22]. Interestingly, the flow cytometry experiments confirmed these findings on the susceptible strains and confirmed that the interaction takes place on the bacterial membrane, which was expected as both molecules act by permeating the outer membrane of Gram-negative bacteria [14]. It is noteworthy that a beneficial effect of the combination was evidenced at low concentrations in the flow cytometry experiment, well below the MICs of the strain studied. This finding corroborates those of previous works that showed the sub-inhibitory effects of antimicrobial peptides [23].

This positive interaction between Pep16 and colistin was reproduced in vivo, with a reduction of bacterial counts in the spleen of mice treated with the combination compared to mice treated with colistin alone, however only on colistin-susceptible *E. coli*. Although it is probably not the only mechanism explaining its increased efficacy, the combination of molecules also reduced the emergence of colistin-resistant mutants in vivo. The beneficial effect of the combination was particularly apparent 6–12 h after the beginning of treatment, which coincides with the peak plasma concentration of Pep16 after subcutaneous administration. Pep16 could therefore allow the use of lower doses of colistin, therefore limiting its toxicity, but also be a partner against some CPEs for which very few antimicrobials remain active. Indeed, some CPEs remain susceptible only to colistin, and the risk of emergence of resistant mutants is then extremely high [22].

In the previous paper by Nicolas et al., Pep16, when used alone, was effective against *S. aureus* in vitro and in mild and severe sepsis mouse models. However, several differences can be identified between their work and ours. First, the MICs of Pep16 for *S. aureus* were significantly lower at 4.5 mg/L, compared with values ranging from 32 to 64 mg/L for *E. coli*. Second, Pep16 was given intravenously in the original study and at different dosages. Third, the model used here, which has a high inoculum and high spontaneous mortality of 97% at 24 h [17], may not have allowed time for Pep16’s activity, considering its delayed peak plasma concentrations.

## 5. Conclusions

Altogether, although less optimistic than for the treatment of Gram-positive organisms and especially *S. aureus* and despite a lack of efficacy when used alone against *E. coli*, Pep16 may be of interest in combination with colistin. Indeed, the combination has proven synergistic in vitro and beneficial in vivo, compared to colistin alone. The precise mechanism of this synergy on the outer bacterial membrane needs to be explored further. It is also of interest that a synergy is evidenced using two compounds that are both derived from peptides and interact with the outer bacterial membrane, whereas synergy is usually described for drugs with distinct mode of actions, suggesting molecular targets on the outer membrane are likely different. The optimal route, dose, and frequency of Pep16 administration, as well as its tolerance also remain to be determined. In the context of the urgent need of new antimicrobials against MDR *Enterobacterales*, Pep16 may potentialize colistin usage by limiting the doses needed to obtain efficacy—thus limiting toxicity—as well as reducing the emergence of colistin-resistant mutants. Furthermore, testing this compound with other partner drugs and other organisms will be necessary and is underway by our group, in order to determine whether it should be pursued as a lead investigational compound and may help fill the therapeutic gap we are facing concerning MDR bacteria in the future.

## Figures and Tables

**Figure 1 antibiotics-12-00081-f001:**
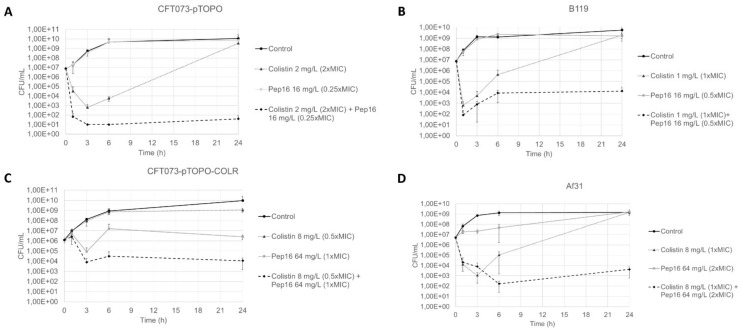
Time–kill curves of Pep16 and colistin alone or in combination against the studied strains at inocula between 10^6^ and 10^7^ CFU/mL. (**A**) CFT073-pTOPO. (**B**) B119. (**C**) Af31. (**D**) CFT073-pTOPO-COLR. Data expressed as the mean and the standard deviation. All experiments were conducted at least 3 times.

**Figure 2 antibiotics-12-00081-f002:**
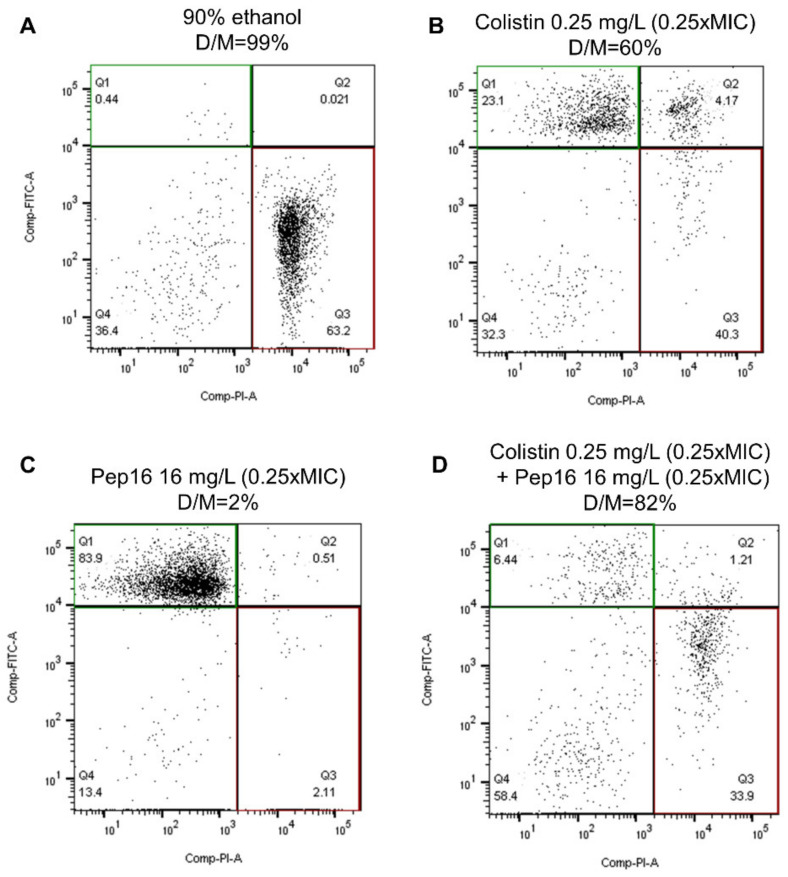
Flow cytometry study of membrane permeability. Bacteria showing green fluorescence have an intact membrane, and bacteria showing red fluorescence have a permeabilized membrane. The percentage of dead/marked is the number of bacteria with red fluorescence divided by the number of bacteria with red and/or green fluorescence. The upper left quadrant (Q1) shows cells with an intact membrane (green fluorescence), and the lower right quadrant (Q3) shows cells with a disrupted membrane. The figure shows the results 30 min after exposure of CFT073-pTOPO to (**A**) 90% ethanol. (**B**) Colistin 0.25 mg/L. (**C**) Pep16 16 mg/L. (**D**) Colistin 0.25 mg/L + Pep16 16 mg/L. D/M: ratio of dead cells/marked cells, calculated as Q3/(Q1 + Q2 + Q3).

**Figure 3 antibiotics-12-00081-f003:**
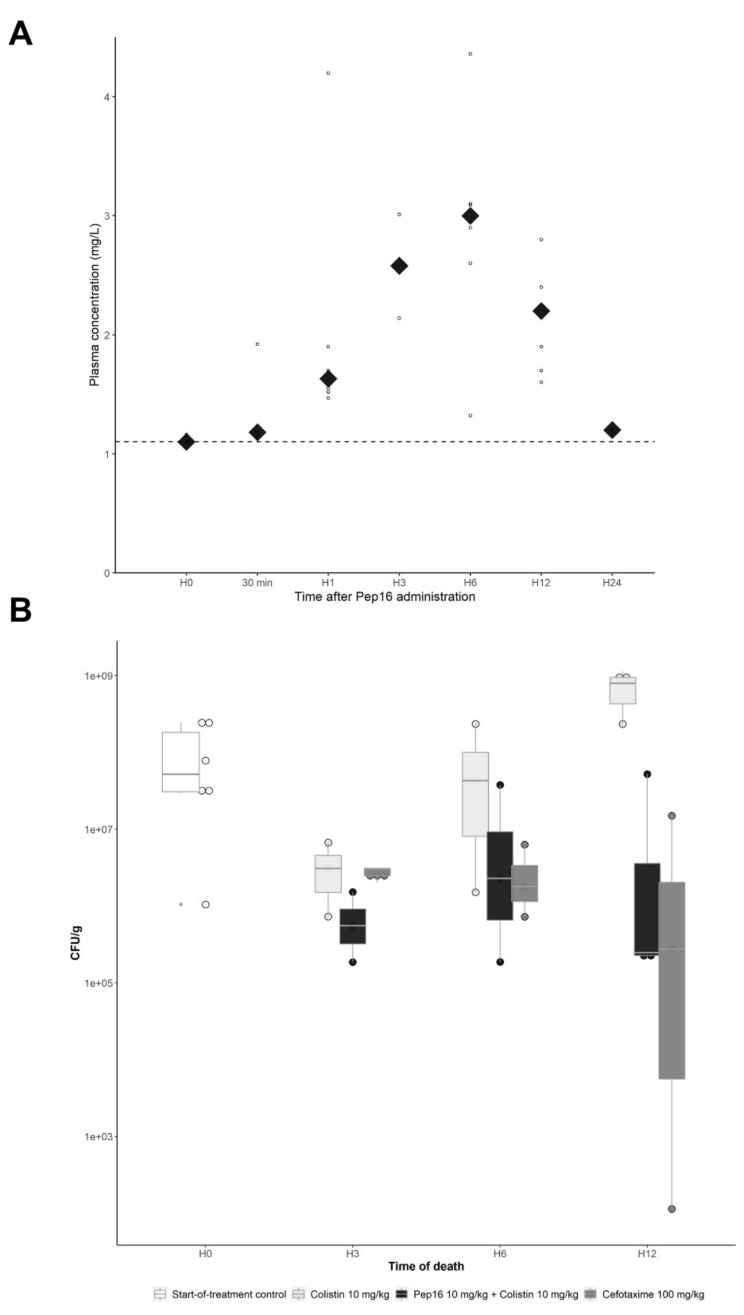
In vivo studies of Pep16. (**A**) Pharmacokinetics of Pep16 in mice after a single subcutaneous administration of 10 mg/kg. Each dot represents a sample, and the diamonds represent the median at each time point. One value at H12 (11.3 mg/L) is not shown on the graph, but has been integrated to determine the median. Dotted line represents the limit of detection. (**B**) Murine model of peritonitis due to CFT073-pTOPO. Bacterial counts in the spleens according to the time of death in 3 treatment groups: colistin alone, colistin + Pep16, and cefotaxime (*p* = 0.002). The group treated with Pep16 10 mg/kg alone is not shown considering the high spontaneous mortality (6/8 mice). Mice that died before their planned time of death were classified as having died at the closest upcoming time point.

**Table 1 antibiotics-12-00081-t001:** MICs of the study strains.

*E. coli* Isolates	Origin	Inoculum (CFU/mL)	MICs		
			Pep16 (mg/L)	Colistin (mg/L) (S ≤ 2; R > 2)	Cefotaxime (mg/L) (S ≤ 1; R > 2)
CFT073-pTOPO	Derived from CFT073 clinical isolate (UTI)	10^5^	64	1	0.06
		10^6^	128	2	1
		10^7^	1024	32	8
CFT073-pTOPO-COLR	In vivo mutant from CFT073	10^5^	64	16	0.06
Af31	Clinical isolate (UTI)	10^5^	32	8	0.06
B119	Clinical isolate (UTI)	10^5^	32	1	0.06

All experiments were performed at least three times. MIC: minimal inhibitory concentration, S: susceptibility, R: resistance. All breakpoints used are those recommended by EUCAST.

## Data Availability

The data presented in this study are available on request from the corresponding author.

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
