# Peer review of "Synergistic Activity of Pep16, a Promising New Antibacterial Pseudopeptide against Multidrug-Resistant Organisms, in Combination with Colistin against Multidrug-Resistant Escherichia coli, In Vitro and in a Murine Peritonitis Model"

_antibiotics, 2023, doi:10.3390/antibiotics12010081_

Round 1

Reviewer 1 Report

Chosidow et al., has investigated pep16 pseudopeptide activity against E. Coli with varying susceptibility of colistin. manuscript is well written and scientifically sounds great. 

I have concern about the Pep16 data is not presented or discussed in the methods sections. How did you get Pep16? whether it is synthesized or purchased? Details should be discussed. 

LC-MS data of pharmacokinetics study must be shown in supporting information.

Is there any analytical data of Pep16 peptide before and after injection into mice? How was the analysis done? 

I recommend this paper for publications in antibiotics after addressing my concern. 

Author Response

Dear Reviewer,

Thank you for the opportunity to revise and re-submit our paper for consideration in the Antibiotics. We appreciate your thoughtful and specific comments , to which we responded to point-by-point, below in bold characters. We believe the revised paper is much improved.

We thank you again for considering our work, 

Sincerely,

Prof. Victoire de Lastours, MD, PhD

Chosidow et al., has investigated pep16 pseudopeptide activity against E. Coli with varying susceptibility of colistin. manuscript is well written and scientifically sounds great. 

I have concern about the Pep16 data is not presented or discussed in the methods sections. How did you get Pep16? whether it is synthesized or purchased? Details should be discussed. 

Thank you for this comment ; We have added in the methods section lines 113 and following:

« Pep16 was supplied by S.A.S. Olgram (Bréhan, France), the manufacturer of the drug, with whom this work was conducted in collaboration. Details on the synthesization of Pep16, inspired by and imitating a section of a Staphylococcus aureus toxin, can be found in the princeps paper published in 2019 [1].

LC-MS data of pharmacokinetics study must be shown in supporting information.

We have added the full results of the phamacokinetics study in the supporting infiormation (File S2).

Is there any analytical data of Pep16 peptide before and after injection into mice? How was the analysis done? 

I’m not sure I undertand this question. We have provided the full dosing data concerning Pep19 after injection in mice (see above).

I recommend this paper for publications in antibiotics after addressing my concern. 

Reviewer 2 Report

Reviewers Comments (Manuscript ID: antibiotics-2111717)

The manuscript by Dr. Lastours et.al. reported the “Synergistic activity of Pep16, a promising new antibacterial pseudopeptide against multidrug resistant organisms, in combination with colistin in a murine Escherichia coli peritonitis model.”. In current article, author describes the synergistic effect of Prp16 with colistin in murine E. coli and it also shows beneficial effect against MDR organism in both in-vitro and in-vivo assay. The current manuscript is well written and activity data are well presented. This reviewer is recommending this manuscript for the publication in antibiotics after incorporating the bellow mirror corrections.

Minor comments:

1.     Abstract can be provided without sub section like introduction, method etc.

2.     Author should provide the structure of Prp16 and colistin in the manuscript.

3.     Table 1 can be move near to first paragraph of results where it cited.

4.     Figure. 1 is too faint. Its quality can be improved.

5.     In page. 11, Figure S1 can be provide in separate file as ESI.

Author Response

Dear Reviewer,

Thank you for the opportunity to revise and re-submit our paper for consideration in Antibioitics. We appreciate the thoughtful and specific comments of the reviewer, to which we responded to point-by-point, below in bold characters. We believe the revised paper is much improved.

We thank you again for considering our work, 

Sincerely,

Prof. Victoire de Lastours, MD, PhD

Internal Medicine Department, Beaujon University Hospital

Assistance Publique Hôpitaux de Paris, Université Paris Cité

IAME Research Group, Université Paris Cité and INSERM, UMR-1137

Reviewer 2

The manuscript by Dr. Lastours et.al. reported the “Synergistic activity of Pep16, a promising new antibacterial pseudopeptide against multidrug resistant organisms, in combination with colistin in a murine Escherichia coli peritonitis model.”. In current article, author describes the synergistic effect of Prp16 with colistin in murine E. coli and it also shows beneficial effect against MDR organism in both in-vitro and in-vivo assay. The current manuscript is well written and activity data are well presented. This reviewer is recommending this manuscript for the publication in antibiotics after incorporating the bellow mirror corrections.

Minor comments:

  1. Abstract can be provided without sub section like introduction, method etc.

This has been done, thank you

  1. Author should provide the structure of Pep16 and colistin in the manuscript.

A reference to the structure of Pep16 and colistin have been added to the manuscript.

  1. Table 1 can be move near to first paragraph of results where it cited.

We leave this to the editing team.

  1. Figure. 1 is too faint. Its quality can be improved.

We have submitted a higher quality version of Figure 1.

  1. In page. 11, Figure S1 can be provide in separate file as ESI.

We leave this to the editing team.

Reviewer 3 Report

The manuscript ‘Synergistic activity of Pep16, a promising new antibacterial pseudo peptide against multidrug-resistant organisms, in combination with colistin in a murine Escherichia coli peritonitis model’ discusses the combinatorial effect of a novel antibacterial pseudo peptide PEP16 combined with colistin by in vitro as well as in vivo studies. The manuscript needs major revision before considering for publication in the current journal. Even though the manuscript is on the synergistic effect of PEP16 and colistin, the authors have not attempted to hypothesize or study the mechanism behind this.

Major comments

The title of the manuscript is not well-defined because it doesn’t reflect the in vitro studies

Line 220: Substantiate why Pep16 did not show any bactericidal effect, despite high concentrations in the discussion part

Discuss the mode of action of Pep16 and colistin and the synergistic effect in detail.

Discuss the colistin resistance mechanism in detail

Table 1 is not referred to well enough in the main manuscript

Are there reports on the effects of PEP16 in combination with other antibiotics? If so, include them

Figure 2 is not discussed well enough in the main manuscript

Minor comments

Expand MBLs in line 43 when you use it for the first time

The method section also has some results explained that could be moved to the results section for better clarity. For instance, lines 118 to 121, 136 to 138, 180 to 185

Increase the font size of the labels and markers in the figure. 1

Mention what the other quadrants signify as well in figure 2

Mention what the brackets signify in table 1

Author Response

Synergistic activity of Pep16, a promising new antibacterial pseudopeptide against multidrug resistant organisms, in combination with colistin against multidrug resistant Escherichia coli, in vitro and in a murine peritonitis model.

Paris, December 23rd 2022

Dear Reviewer,

Thank you for the opportunity to revise and re-submit our paper for consideration in Antibioitics. We appreciate the thoughtful and specific comments of the reviewer, to which we responded to point-by-point, below in bold characters. We believe the revised paper is much improved.

We thank you again for considering our work, 

Sincerely,

Prof. Victoire de Lastours, MD, PhD

Internal Medicine Department, Beaujon University Hospital

Assistance Publique Hôpitaux de Paris, Université Paris Cité

IAME Research Group, Université Paris Cité and INSERM, UMR-1137

Reviewer 3

The manuscript ‘Synergistic activity of Pep16, a promising new antibacterial pseudo peptide against multidrug-resistant organisms, in combination with colistin in a murine Escherichia coli peritonitis model’ discusses the combinatorial effect of a novel antibacterial pseudo peptide PEP16 combined with colistin by in vitro as well as in vivo studies. The manuscript needs major revision before considering for publication in the current journal. Even though the manuscript is on the synergistic effect of PEP16 and colistin, the authors have not attempted to hypothesize or study the mechanism behind this.

Major comments

The title of the manuscript is not well-defined because it doesn’t reflect the in vitro studies

Thank you for this comment. We have changed the title to :

“Synergistic activity of Pep16, a promising new antibacterial pseudopeptide against multidrug resistant organisms, in combination with colistin against multidrug resistant Escherichia coli, in vitro and in a murine peritonitis model.”

Line 220: Substantiate why Pep16 did not show any bactericidal effect, despite high concentrations in the discussion part.

Discuss the mode of action of Pep16 and colistin and the synergistic effect in detail.

The mechanism of action of the Pep16 peptide (as most antibacterial peptides) is incompletely elucidated. Studies using electron microscopy of E. coli and Staphylococcus aureus after incubation with Pep16 showed alterations of the bacterial wall. Membrane permeability studies confirmed the penetration of Pep16 and the disruption of membrane integrity by Pep16. Additionnally, Nuclear Magnetic Resonance study suggested interactions between the hydrophobic chains of these peptides and the bacterial membrane (Nicolas et al., 2019).

Colistin displaces the divalent cations of calcium (Ca2+) and magnesium (Mg2+) in a competitively way, impairing the LPS three-dimensional structure. Colistin then inserts its hydrophobic terminal acyl fat chain, causing an expansion of the external outer membrane (OM) monolayer. A permeabilization of OM occurs, allowing colistin to get through OM as a self-promotion (Andrade et al, 2020).

We hypothesized that combining Pep16 and colistin which both operate on the outer membrane of Gram-negative bacteria - may potentiate their antibacterial effect. Colistin, indeed, is In the study of membrane permeability on CFT073-pTOPO, the percentage of dead cells/marked cells after exposure to Pep16 0.25xMIC, to colistin 0.25xMIC and to the combination of both molecules was 2%, 60% and 82% (p<0.001), respectively (Fig 2). In the absence of a validated definition of synergy in flow cytometry experiments, the increase in the percentage of dead cells/marked cells with the combination of Pep16 and colistin suggests a beneficial effect of the combination against outer bacterial cell membrane.

References :

Ferdinando F. Andrade, Daniela Silva, Acácio Rodrigues, and Cidália Pina-Vaz.Colistin. Update on Its Mechanism of Action and Resistance, Present and Future Challenges ; Microorganisms. 2020 Nov; 8(11): 1716. doi: 10.3390/microorganisms8111716

Nicolas, I.; Bordeau, V.; Bondon, A.; Baudy-Floc’h, M.; Felden, B. Novel Antibiotics Effective against Gram-Positive and -Negative Multi-Resistant Bacteria with Limited Resistance. PLoS Biol. 2019, 17, e3000337, doi:10.1371/journal.pbio.3000337.

Discuss the colistin resistance mechanism in detail.

Two colistin resisatnt strains were used, as described in the methods section. The first had been selected from  CFT073 under treatment with colistin in the peritoneal fluid from mice using the same murine peritonitis model (Fantin et al, 2019) . The mechanism of resistance was a chromosomic mutation. The second was a clinical strain isolated in a patient’s urine. This strain harbored the mcr-1- plasmid that was responsible for colistin resistance (E. coli Af31). The details of these strains and resistance mechanims are to be found in the article referenced in the methods section.

Fantin, B.; Poujade, J.; Grégoire, N.; Chau, F.; Roujansky, A.; Kieffer, N.; Berleur, M.; Couet, W.; Nordmann, P. The Inoculum Effect of Escherichia Coli Expressing Mcr-1 or Not on Colistin Activity in a Murine Model of Peritonitis. Clin. Microbiol. Infect. Off. Publ. Eur. Soc. Clin. Microbiol. Infect. Dis. 2019, 25, 1563.e5-1563.e8, doi:10.1016/j.cmi.2019.08.021.

Table 1 is not referred to well enough in the main manuscript

We have referred table 1 in the manuscript in the results section.

Are there reports on the effects of PEP16 in combination with other antibiotics? If so, include them

No data on Pep16 associated with other antibiotics exist to date.

Figure 2 is not discussed well enough in the main manuscript

We have added line 235 : « Figure 2 shows the results of the flow cytometry experiments used to investigate membrane permeability after exposure to antibiotics”

Minor comments

Expand MBLs in line 43 when you use it for the first time

This has been done.

The method section also has some results explained that could be moved to the results section for better clarity. For instance, lines 118 to 121, 136 to 138, 180 to 185

We have moved lines 180 down to the results section, lines 253 and following :

“Of note, pharmacokinetic analysis of colistin and cefotaxime had already been completed and were not repeated for ethical reasons [3,21]. »

Increase the font size of the labels and markers in the figure. 1

This has been done.

Mention what the other quadrants signify as well in figure 2

The legend of figure 2 has been changed for clarity :

“Fig 2: Flow cytometry study of membrane permeability. Bacteria showing green fluorescence have an intact membrane, bacteria showing red fluorescence have a permeabilized membrane. The percentage dead cells/marked cells (or D/M ratio) is the number of bacteria with red fluorescence divided by the number of bacteria with red and/or green fluorescence. The upper left quandrant (Q1) shows cells with n intact membrane (green fluorescence), the lower right quadrant (Q3) shows cells with a disrupted membrane. The figure shows the results 30 min after exposure of CFT073-pTOPO to (A) 90% ethanol. (B) Colistin 0.25 mg/L. (C) Pep16 16 mg/L. (D) Colistin 0.25 mg/L + Pep16 16 mg/L”

D/M: ratio of dead cells/marked cells, calculated as Q3/(Q1+Q2+Q3)

Mention what the brackets signify in table 1

The brackets have been deleted for clarity.